# Venom Function of a New Species of *Megalomyrmex* Forel, 1885 (Hymenoptera: Formicidae)

**DOI:** 10.3390/toxins12110679

**Published:** 2020-10-29

**Authors:** Kyle Sozanski, Lívia Pires do Prado, Andrew J. Mularo, Victoria A. Sadowski, Tappey H. Jones, Rachelle M. M. Adams

**Affiliations:** 1Department of Evolution, Ecology and Organismal Biology at The Ohio State University, Columbus, OH 43210, USA; sozanski.1@osu.edu (K.S.); sadowski.41@osu.edu (V.A.S.); 2Coordenação de Ciências da Terra e Ecologia, Museu Paraense Emílio Goeldi 66077-830, PA, Brazil; livia.pires7@gmail.com; 3Department of Biological Sciences, Purdue University, Indiana, IN 47907, USA; amularo@purdue.edu; 4Department of Chemistry, Virginia Military Institute, Virgina, VA 24450, USA; JonesTH@vmi.edu; 5Department of Entomology, Smithsonian Institution, National Museum of Natural History, Washington, DC 20560, USA

**Keywords:** venom, alkaloid, *Megalomyrmex peetersi*, insecticide, antibiotic

## Abstract

Alkaloids are important metabolites found across a variety of organisms with diverse ecological functions. Of particular interest are alkaloids found in ants, organisms well known for dominating the ecosystems they dwell in. Within ants, alkaloids are found in venom and function as potent weapons against heterospecific species. However, research is often limited to pest species or species with parasitic lifestyles and thus fails to address the broader ecological function of ant venom alkaloids. Here we describe a new species of free-living *Megalomyrmex* ant: *Megalomyrmex peetersi* sp. n. In addition, we identify its singular venom alkaloid (*trans*-2-butyl-5-heptylpyrrolidine) and elucidate the antibiotic and insecticidal functions of its venom. Our results show that *Megalomyrmex peetersi* sp. n. venom is an effective antibiotic and insecticide. These results are comparable to venom alkaloids found in other ant species, such as *Solenopsis invicta.* This research provides great insight into venom alkaloid function, and it is the first study to explore these ideas in the *Megalomyrmex* system.

## 1. Introduction

From nicotine to serotonin, alkaloids are a diverse group of metabolites active across many forms of life with far-reaching ecological ramifications [1]. Alkaloids are loosely defined as naturally occurring heterocyclic organic compounds containing a nitrogenous base in a negative oxidation state. They are classified based on biological activity, taxon of discovery, and chemical structure and are used in various capacities [1]. The most well-known alkaloids are those synthesized by plants, such as caffeine by coffee and nicotine by tobacco, as a method of defense against herbivory [2,3]. Approximately 25% of higher plants synthesize unique alkaloids that serve similar functions [1]. Endophytic fungi, found within plants, also produce alkaloids that are toxic to herbivorous insects and microbes [4], thus serving as a protective symbiont [5]. Despite alkaloids being toxic to many insect herbivores, some have evolved the ability to sequester them after ingestion [6], employ them in personal defense [7,8], and even allocate them to offspring for protection [9]. Insectivorous predators can also sequester alkaloids for protection by ingesting alkaloid-synthesizing prey (e.g., ants, millipedes, mites), with one of the most notable examples being the conspicuously colored poison dart frogs (Dendrobatidae) [10]. In these ways, alkaloids are powerful drivers of ecological interactions. This is perfectly exemplified in ants, a group of organisms well known for defining the ecosystems they dwell in [11].

Alkaloids are rare among the ca. 15,000 ant species, only found in approximately 10 of the 500 genera [12,13]. Eight of these genera belong to the widely studied tribe Solenopsidini, where alkaloids are venom components [13]. *Solenopsis* and *Monomorium,* the most well-studied genera of the Solenopsidini tribe, employ pyrrolidine and piperidine alkaloids communicatively and combatively towards heterospecific species. Two pyrrolidine venom alkaloids are utilized by the *Monomorium rothsteini* species complex in interspecific competition to repel other ant species from food, which allows for a monopolization of resources [14,15]. *Solenopsis fugax* (Latreille), a facultative parasite of other ants, also deploys a repulsive pyrrolidine when invading host nests to steal brood. Perhaps most notably, piperidines are responsible for the “burning” sensation that the fire ant species *Solenopsis invicta* Buren and *Solenopsis geminata* (Fabricius) deliver when they sting, creating an effective defense against threats to the colony [16]. These same *Solenopsis invicta* piperidines exhibit strong insecticidal properties [17], allowing *S. invicta* to be an effective predator and outcompete other ant species [16]. In addition to these combative properties, *S. invicta* alkaloidal venom has been shown to have antibacterial properties, inhibiting both Gram-positive and Gram-negative bacteria [18]. *Solenopsis invicta* queens utilize these antibacterial properties by spreading venom onto their eggs, inhibiting the growth of pathogenic microbes [19]. *Solenopsis invicta* workers, which also produce these alkaloids, have been speculated to disperse their venom onto offspring through gaster flagging within the brood chambers as antimicrobial protective agents [20]. However, focus on *Solenopsis* and *Monomorium* fails to fully capture ant alkaloid diversity and thus limits our understanding of the ecological implications of ant alkaloids. Among the alkaloid-synthesizing ant genera, *Megalomyrmex* has great alkaloid structural diversity [21,22,23,24,25,26], offering an opportunity to further elucidate ant alkaloid evolution and function.

The neotropical genus *Megalomyrmex* (Solenopsidini) consists of 44 species that produce at least five classes of alkaloid: pyrrolidines, pyrrolizidines, piperidines, pyrrolines, and indolizidines [21,22,23,24,25,26]. This alkaloid structural diversity is complemented with variation in lifestyle and natural history traits influenced by venom use. Approximately 10 species are social parasites (one social species parasitizing another) that span a spectrum of lifestyles from facultative thieves to obligate guest ants [21,23]. The thief ants *Megalomyrmex mondabora* Brandão*, Megalomyrmex mondaboroides* Longino, and *Megalomyrmex silvestrii* Wheeler produce eight different alkaloids from three different structural classes and use these compounds to subdue their fungus-farming ant hosts by stinging or emitting an alkaloidal aerosol [23]. The guest ants *Megalomyrmex adamsae* Longino use their venom against host queens while infiltrating the nest [26]. Similarly, *Megalomyrmex symmetochus* Wheeler guest ants use their venom to suppress host aggression but, remarkably, also use their venom against another competing social parasite, *Gnamptogenys hartmani* (Wheeler) raider ants. *Megalomyrmex symmetochus* venom serves not only as a lethal toxin to these raiders but also as a behavior modifier, causing the raider ants to turn on one another and kill their own nestmates [22]. The majority of *Megalomyrmex* are free-living, nonparasitic species that produce a wide array of pyrrolidine alkaloids [26], likely used as repellents during competitive interactions when scavenging [24]. One example of the efficacy of free-living *Megalomyrmex* alkaloids comes from the new species *Megalomyrmex peetersi* we describe here. While observing ant baits in Costa Rica, we found that a few workers of the new species *Megalomyrmex peetersi* (referred to as *Megalomyrmex wallacei* Mann) arriving to a cookie bait monopolized by *Pheidole* workers were able to disperse hundreds of individuals by exhibiting characteristic alkaloid-dispending behaviors known as gaster flagging [20,23] and “bucking” behavior [27].

Here we describe the new free-living species *Megalomyrmex peetersi*, which synthesizes *trans*-2-butyl-5-heptylpyrrolidine as its exclusive venom alkaloid. This alkaloid has previously been found in several *Monomorium* Mayr species [28,29], *Solenopsis* Westwood species [15,30], and other *Megalomyrmex* Forel species [24,25,31]. Given that *M. peetersi* sp. n. is a free-living ant species that nests in the leaflitter and likely interacts with microbial pathogens and other insect species [32], we sought to investigate the antimicrobial and insecticidal properties of its venom. This is among the first studies to experimentally describe alkaloidal function in free-living *Megalomyrmex,* furthering our understanding of this genus. More broadly, this work will provide greater insight into how alkaloids influence ecological interactions in an important group of organisms.

## 2. Results

### 2.1. Taxonomic Account

*Megalomyrmex peetersi* Prado and Adams, sp. n. zoobank.org:act:B335D580-B01B-48F8-8AC9-C906D6E402C0 *Megalomyrmex wallacei* Mann, 1916: 445 (in part).

#### 2.1.1. Repositories

Specimens from the following myrmecological collections were studied and are stored at: Coleção Entomológica Padre Jesus Santiago Moure, Universidade Federal do Paraná (DZUP; Curitiba, PR, Brazil); John Longino Personal Collection, University of Utah (JTLC; Salt Lake City, UT, USA); Coleção Entomológica do Museu Paraense Emílio Goeldi (MPEG; Belém, PA, Brazil); Coleção de Hymenoptera do Museu de Zoologia da Universidade de São Paulo (MZSP; São Paulo, SP, Brazil); and Museum of Comparative Zoology (MCZ; Cambridge, MA, USA).

#### 2.1.2. Type Material

Holotype worker (Figure 1). Costa Rica: Heredia: La Selva Biological Station, 50–150 m, 10°26′ N 84°01′ W, iv.1994, INBio-OET, N Barger & J Longino cols., baiting study, NNB/PLT/02, INBIOCRI001242851 MZSP.

Paratype workers (6 workers). Same data as holotype: INBIOCRI001242849 MZSP, INBIOCRI001242850 MZSP, INBIOCRI001242854 MZSP, INBIOCRI001242856 MZSP, INBIOCRI001242857 DZUP, INBIOCRI001242858 MPEG.

#### 2.1.3. Diagnosis (Worker)

Piligerous punctures on head surface slightly raised and coarser than mesosoma (Figure 1a)Dental formula 1 + 4 equally spaced, with the first tooth slightly smaller and the apical tooth slightly larger (Figure 1b)Antennal clava three segmented comparatively enlarged (Figure 1c)Promesonotal suture distinct and well impressed (Figure 1d)Metanotal sulcus deeply impressed and wideAbsence of a sulcus between the anepisternum and katepisternumVentral portion of the petiole with a translucent flangeIn lateral view, ventral region of postpetiole with globose shape (Figure 1e)

#### 2.1.4. Worker Description

Measurements (*n* = 6, holotype values within parentheses): HW 0.96–1.03 (0.96), HL 1.32–1.46 (1.32), ML 0.59–0.71 (0.61), EL 0.30–0.36 (0.34), SL 1.53–1.75 (1.53), WL 1.78–2.05 (1.92), PrW 0.69–0.75 (0.69), MFL 1.84–2.13 (1.84), PL 0.75–0.8 (0.76), PH 0.48–0.53 (0.48), PPL 0.48–0.71 (0.48), PPH 0.48–0.51 (0.5), ATW 1.09–1.26 (1.09). 

Worker (Figure 1). *Color.* Surface of the general body ranging from orange-brown to dark brown; when orange-brown, the head and gaster slightly darker (Figure 1d,f); pilosity yellowish. *Pilosity.* Filiform, suberect, and abundant, evenly distributed along the entire body surface. Ocular pilosity present, with sparse flexuous setae. Antennal clava with appressed pilosity, relatively smaller and thinner from the remainder of the antennae. Anterior margin of the clypeus with sparse and elongated pilosity, especially those of the median portion, which is even more elongated. Apex of tarsomeres inner face with usually three or more robust spiniform setae. *Surface sculpturing.* Shiny appearance with piligerous punctures usually coarser on the head. Surface of mandibles smooth and shiny, with abundant piligerous punctuations (Figure 1b); frontal region of the head, just below the eyes with concentric carinae, ranging in quantity and degree of impression; remaining body predominantly smooth; except for the presence of the two longitudinal carinae crossing the region of the metapleural gland bulla, for the propodeal declivity with few concentric carinae, and apical portion of the gaster, finely imbricate.

*Size.* Medium (WL 1.78–2.05). *Head.* Oval shaped with distinct occipital carina, visible in frontal view (Figure 1a). Dental formula 1 + 4 equally spaced, with the first tooth slightly smaller and the apical tooth slightly larger (Figure 1b). Median portion of clypeus broad and rounded. Frontal carina very short, not extending beyond the posterior margin of the antennal fossa. Antennal clava three segmented comparatively enlarged (Figure 1c), apical segment slightly longer than the second and third. Eyes round, larger, protruding, with about 15 small ommatidia in diameter, diagonally arranged placed just below midlength of the head. *Mesosoma.* In lateral view, mesosoma forming two convexities, a slightly higher formed by promesonotum and a lower formed by propodeum (Figure 1d). Promesonotal suture well impressed and metanotal groove deep and wide; absence of a sulcus between the anepisternum and katepisternum. Propodeal spiracle round, surrounded by the cuticular swell. Foraminal carina complete and well developed. *Metasoma.* In lateral view, subpedunculated petiole, with petiolar node subtriangular; ventral portion of the petiole with a narrow translucent flange. In lateral view, region anterior and posterior of postpetiole with globose shape (Figure 1e), without projections.

#### 2.1.5. Ergatoid Queen

Measurements *(*n = 1): HW 1, HL 1.36, ML 0.67, EL 0.36, SL 1.53, WL 1.94, PrW 0.78, MFL 1.84, PL 0.71, PH 0.55, PPL 0.53, PPH 0.55, ATW 1.42.

Comments (Figure 2). Morphologically similar to the worker, distinguished by: *Pilosity*. Erect and abundant, distributed along the entire body surface (Figure 2a–c). *Surface sculpturing.* Piligerous punctures coarser and distributed over the entire body surface; concentric carinae present in the frontal lobes and in the propodeum are coarser and abundant. *General body.* Size slightly larger and gaster substantially higher. Teeth more pointed. Lateral ocelli present and median ocellus absent. Promesonotum strongly convex; promesonotal suture strongly impressed (Figure 2c). In lateral view, petiolar node thinner; ventral portion of postpetiole more pronounced (Figure 2c).

#### 2.1.6. Male

Measurements (n = 1): HW 0.97, HL 1.19, ML 0.59, EL 0.57, SL 0.57, WL 2.42, PrW 1.04, MFL 1.85, PL 0.92, PH 0.47, PPL 0.66, PPH 0.59, ATW 1.38.

Comments (Figure 3). Boudinot et al. [28] described the male of this species as *M. wallacei*. According to the authors and with the specimen examined in this work, males of this species are morphologically recognized by (1) the presence of blackened piligerous punctures on the body surface (Figure 3a,b); (2) third antennal segment apically flattened and curved (Figure 3c); (3) in lateral view, petiole with raised and convex node and postpetiolar node globose dorsally (Figure 3b); (4) ventral region of postpetiole conspicuous, as inverted triangle (Figure 3b).

#### 2.1.7. Distribution and Nesting Biology

*Megalomyrmex peetersi* has been recorded from South America and Central America (Figure 4). In South America, it was recorded in the departments of Chocó and Antioquia. In Central America, the species has been found in protected areas of lowland tropical rainforests in Costa Rica and Panama.

All colonies were found nesting either close to the ground in vegetation or within the leaflitter. When on or near live plants, they were at the base of palm leaves or bromeliads. They most frequently occupied old fallen palm stems with hollowed cavities ca. 5–10 cm in length. Signs suggest the ants manipulate nesting material and structure by adding debris to cover holes or build thin walls. When nesting between leaves on the ground, workers hang upside down on the top leaf holding brood. Colonies readily move with the slightest disturbance and, therefore, need to be collected quickly after discovery.

Palm vegetations (Arecaceae) identified in Costa Rica include *Welfia regia*, *Cryosophila warscewiczii*, *Asterogyne martiana*, and *Geonoma* spp. Other vegetations include *Dipteryx panamensis* trees, *Psychotria elata* shrubs, and *Salpichlaena volubilis* ferns, as well as *Asplundia* spp., *Piper* spp., *Adiantum* sp., *Aechmea* sp., *Rinorea deflexiflora, Anthurium* spp., *Spathiphyllum* sp., and *Philodendron* sp.

#### 2.1.8. Additional Information

##### Etymology

The species was named in honor of the myrmecologist Dr. Christian Peeters (1956–2020). Dr. Peeters was born in Belgium and, since 1991, has served as a research professor at the Institute of Ecology and Environmental Sciences at Sorbonne University, France. He made valuable contributions to science as a teacher, advisor, researcher, naturalist, and science communicator. Throughout his brilliant trajectory, he combined field and laboratory work, publishing important contributions mainly in the study of the evolutionary divergence of castes and the strategies of colony foundation. All authors of this paper regret his early departure.

In addition, Peeters and Adams studied the reproductive behavior of this species (considered *M. wallacei* in the paper) [33]. In this work, the authors recommended a more in-depth study of the populations in Brazil and Central America, suggesting that it could be a new species.

##### Comments

*Megalomyrmex peetersi* was previously identified as *Megalomyrmex wallacei* [27,34,35] (e.g., Appendix A). However, Boudinot et al. [27] and Peeters and Adams [33] commented on the differences between specimens from South America and Central America, indicating the need for a more comprehensive study. In the taxonomic revision in preparation [36], studies of the *M. wallacei* type material deposited at the MCZ and an increase in sampling in the Brazilian Amazon revealed several differences between species (Appendix A), which include (1) protuberance of the eyes and number of ommatidia (in diameter), with about 15 in *M. peetersi* and 20 in *M. wallacei*; (2) absence of a sulcus between the anepisternum and katepisternum in *M. peetersi;* (3) shape of the propodeum in lateral view, weakly convex in *M. peetersi* (Figure 2b) and plane in *M. wallacei*; (4) in *M. peetersi*, ventral portion of the postpetiole globose (as mentioned by Boudinot et al. [27]) (Figure 2) and irregular and weakly convex in *M. wallacei*; and (5) the surface of the integument smooth and shiny in *M. peetersi* and with subopaque appearance and the presence of a stronger sculpturation on the surface of the head, mesosoma, and waist in *M. wallacei* (Appendix A). Finally, beyond morphology, the distribution and behavior differences reinforce the argument for the species distinction. While *M. wallacei* is distributed throughout the Amazon and in the Amazon–Cerrado transition zone, *M. peetersi* is distributed in Central America and northwestern Colombia (Figure 4). Regarding the reproductive strategy, as reported by Peeters and Adams [34], *M. peetersi* has only ergatoid queens performing the reproductive function in the colony, while *M. wallacei* is polygynous, with the presence of true queens in the colony.

In this sense, in the latest taxonomic revisions for *Megalomyrmex* [27,35], with the exception of the specimen from Guyana [35] *M. wallacei* is now considered as *M. peetersi*.

##### Additional Material Examined

COLOMBIA: Antioquia: Amalfi, Cañon del Porce, Normandia, 1045 m alt., 6°57′01″ N 75°11′36″ W, 19.xii.1999, E. Vergara & F. Serna cols., En bosque, Winkler (3 workers, MZSP). Chocó: Parque Nacional Natural Utría, Ensenada, 06°01′01″ N 77°20′55″ W, bosque abierto, 21.v.1991, M. Baena leg (2 workers, MZSP). COSTA RICA: Heredia: La Selva Biological Station, 50–150 m, 10°26′ N 84°01′ W, iv.1994, N. Barger & J. Longino cols., baiting study, NNB/PLT/02, INBio-OET, INBIOCRI001242855 (worker, JTLC). Same data except 17.i.1993, INBIOCRI001223542 [male, MZSP]. Same data except J. Longino col., #3733, INBIOCRI001254263 (worker, MZSP). Same data except 10°25′35.04″ N 84°1′32.88″ W, 93 m, 19.viii.2003, Rachelle M.M. Adams col., #RMMA030819-07 (worker, MZSP); #RMMA030819-08 (worker, MZSP); CASENT0630979 (worker, JTLC); RMMA030819-10 (worker and ergatoid queen, MZSP). Same data except 10.vii.2005, #RMMA050710-2, CASENT0630962 (2 workers, MZSP; ergatoid queen, JTLC). Same data except 83°59′ W 10°26′ N, 5.vii.2005, RMM Adams col., RMMA050703-081 (ergatoid queen, MZSP). PANAMA: Chepo, El Llano, CRC180623-08 (9 workers, MPEG). Same data except CRC180623-09 (worker, MPEG). Same data except R. Adams cols., RMMA180623-18 (3 workers, MPEG). Total: 26 workers, 3 ergatoid queens, and 1 male.

### 2.2. Chemical Analysis

Gas chromatography–mass spectrometry (GC–MS) was performed on four workers and four larvae from two ant colonies (RMMA180623-18 and CRC180623-09) collected near El Llano, Panama (GPS). Single ants and two larvae were submerged and extracted in methanol. The GC–MS was set to EI mode, using a Shimadzu QP-2010 GC–MS or a Shimadzu QP-2020 GC–MS with an RTX-5, 30 m × 0.25 mm i.d. column. This device was set for analysis from 60 to 250 °C at 10°/min. The alkaloid in the extracts of *M. peetersi* had identical mass spectrum and gas chromatographic retention time with those of the second eluting isomer of an authentic sample of 2-butyl-5-heptylpyrrolidine.

### 2.3. MIC Assays

A reduction in growth was observed for all six tested bacterial strains as the concentration of 2-butyl-5-heptylpyrrolidine was increased. Bacterial growth was inhibited, as indicated by a statistically significant decrease in OD_600_ at mid-log phase compared with EtOH at concentrations of 31.25–62.5 μg/mL for all bacterial strains (Table 1).

No difference in sensitivity was found between Gram-positive and Gram-negative strains (Figure 5). The inhibitory effect of 2-butyl-5-heptylpyrrolidine is comparable to the broad-spectrum bacteriostatic antibiotic tetracycline at a biologically relevant concentration (Figure 5).

Raw venom also appears to inhibit bacterial growth at increasing concentration using the same broth microdilution method, although differences in OD_600_ from the EtOH control were not statistically significant. This could be due to unknown interactions between raw venom, EtOH, and other factors.

### 2.4. Insecticidal Assays

Immediately following venom application, some termites reacted by gnashing their mandibles, “flinching,” or secreting oral fluids. Termites often showed “intoxication” symptoms following contact with *M. peetersi* venom. These symptoms consistently arose prior to death and lasted at various amounts of time. When affected, termites would follow a pattern of loss of motor control, followed by paralysis, then death. We focused on quantifying the insecticidal properties of ant venom by measuring mortality over time (Figure 6). A total of 62% of termites died after 1 h of venom application, and 72% of termites were dead after 6 hours. These results suggest that *M. peetersi* venom is a potent insecticide.

### 2.5. LD_50_ Assays of Synthetic Alkaloid

Mean termite weight was 2.65 mg. LD_50_ for the alkaloid was calculated to be 5.21 μg/mg with a standard error of 26.693. This high error rate can be explained by insufficient mortality, possibly due to small dose amounts.

## 3. Discussion

Alkaloids are a broad class of chemicals found across disparate forms of life. These compounds have far-reaching ecological ramifications, not only for the species that synthesize them, but also for those that interact with alkaloid-synthesizing species. Despite the compound diversity, research exploring alkaloid function is lacking and disproportionally describes their defensive functions [1]. In particular, little is known of the functions of alkaloids in ants, some of the most ecologically impactful and chemically diverse organisms. *Megalomyrmex* species offer an opportunity to address this gap, given the combination of multiple lifestyles and the variety of alkaloids they synthesize compared with other ants. This study is the first to address alkaloid function in a free-living species of *Megalomyrmex* ant. Our results describe a new species of *Megalomyrmex* ant, identifying the alkaloid it synthesizes and detailing the likely functions of this alkaloid.

We confirmed with an authentic sample that the exclusive alkaloid of *Megalomyrmex peetersi* is *trans*-2-butyl-5-heptylpyrrolidine, which is also found in the venom of multiple species of Solenopsidini ants [15,25,28]. Research of Solenopsidini venom addresses several different topics, but we focused on possible antibiotic and insecticidal functions, given that *M. peetersi* is a soil-dwelling ant species likely to interact with microbial pathogens and other insect species through predation or competition. 

Here we show that 2-butyl-5-heptylpyrrolidine serves as an effective broad-spectrum antibacterial compound at low concentrations. The minimum inhibitory concentration of the synthetic alkaloid ranged from 31.25 to 62.5 μg/mL across six bacterial strains. Our results are comparable to other studies, where pyrrolidine alkaloids bgugaine and irniine found in the plant *Arisarum vulgare* Tozzetti also inhibited Gram-positive bacteria at low concentrations (6.25–50 μg/mL) [37]. When compared with halogenobenzene pyrrolidine and piperidine derivatives, many of these compounds are effective antimicrobials at concentrations higher than that of 2-butyl-5-heptylpyrrolidine (e.g., 256–512 μg/mL) [38]. However, future studies on a wider range of microorganisms are needed to resolve the mechanism of action for the antimicrobial properties of this compound. 

The compound 2-butyl-5-heptylpyrrolidine was found on the cuticle of larvae as well as within the venom reservoir of mature workers, suggesting that workers may apply this venom alkaloid to their brood. Workers of other Solenopsidini species are known to apply alkaloids to the soil walls of their nest in response to pathogens or for use as a prophylactic (e.g., [32]). It is possible that *M. peetersi* applies its alkaloid venom to brood for similar reasons. Further research is needed to establish the ecological role of venom application behavior in *M. peetersi* and other free-living predatory ants.

Our results also show that 2-butyl-5-heptylpyrrolidine serves as an effective insecticide, with both toxic and behavior-modifying qualities. Our results are comparable to pyrrolidine alkaloids found in *Monomorium monomorium* Bolton, where LD_50_ varied from 0.11 to 3.54 μg/mg in termite workers [39]. Fox et al. [17] reported intoxication effects resulting from *Solenopsis*
*invicta* and *S. geminata* pyrroline venom alkaloids with similar descriptions to what we observed with *Megalomyrmex peetersi* venom. Previous studies in ant venom alkaloids have proposed these effects to be a result of alkaloids acting on the central nervous system [39], although these interactions are still unclear in vivo. Nicotine, which is well known for its insecticidal function, exhibits similar toxic and behavior-modifying qualities [1,3]. These results, combined with observations of *Megalomyrmex peetersi* workers applying venom to offered prey items during feeding, reinforce *trans*-2-butyl-5-heptylpyrrolidine’s role as an effective tool when hunting prey.

The singular venom alkaloid of *Megalomyrmex peetersi* functions as an effective antibacterial and insecticidal agent. Paired with their natural history as a leaflitter-dwelling predator, we propose that *M. peetersi* leverages these functions to thrive in its environment. These results provide insight not only for this species but for the broader ecological functions of ant alkaloids, particularly those in the *Megalomyrmex* genus with its unparalleled alkaloid diversity. Questions remain regarding functions of other structurally distinct ant alkaloids, how certain functions may be interconnected (e.g., if insecticidal function enables resource domination), and details concerning in situ alkaloid application.

## 4. Materials and Methods

### 4.1. Insect Colonies

Live *Megalomyrmex peetersi* colonies were collected and observed by Rachelle M.M. Adams (August 2003, *n* = 5; August 2005, *n* = 11; and March 2011, *n* = 12) from La Selva Biological Station (10°24′59″ N, 084°01′12″ W, 50 m elevation) [28,31]. In June 2018, two live colonies were collected from a new site near El Llano, Panama (El Llano forest, 9°16′46.40″ N 78°57′41.40″ W, 365 m) (RMMA180623-18 and CRC180623-09). Live colonies were transported in temporary petri dishes or small plastic containers and then kept in ant-rearing facilities at the University of Texas at Austin, the University of Copenhagen, and the Museum of Biological Diversity at the Ohio State University. Panamanian colonies were placed in a darkened wooden cabinet (ca. 23 °C). Ant enclosures consisted of multiple containers lined with plaster of Paris™, as well as tubes and petri dishes lined with moistened cotton, which provided ample nesting choices for the ants. Containers lined with plaster of Paris™ were watered 2–3 times a week, and cotton from the tubes and dishes were rehydrated and changed as necessary. A diet of Bhatkar agar (consisting of water, honey, eggs, agar, Wesson salts™, Vanderzants vitamin mix™) and live flightless *Drosophila hydei* Sturtevant and *Drosophila melanogaster* Meigen fruit flies were provided 3 times a week, which were both readily consumed by workers.

Five mature queen- and king-right colonies of the termite species *Reticulitermes flavipes* Kollar collected near Columbus, Ohio (39°57′40.3194″ N 82°59′55.68″ W), were used in insecticidal activity assays. Termites were housed inside large 10 × 16 inch plastic containers and fed a mixture of wood mulch and blocks to consume and nest in. Water was applied every few days unless condensation had formed on the container. Colonies were reared in a controlled insect-rearing room (ca. 23 °C).

### 4.2. Taxonomic Account

#### 4.2.1. Terminology

The morphological terminology used follows Wilson [40] and Esteves and Fisher [41] for pilosity, Harris [42] for surface sculpture, and Bolton [43], Longino [35], and Boudinot et al. [27] for the overall external morphology.

#### 4.2.2. Measurements 

Measurements employed were made at 50× magnifications with a Leica MZ7.5 stereomicroscope. The measurements are given in millimeters, and abbreviations are detailed below:HW. *Head width*. Maximum width of head in full-face view (excluding the eyes).HL. *Head length.* Maximum length of head in full-face view from anterior margin of clypeus to posterior margin of head, including occipital carina [27].ML. *Mandibular length.* Straight length of mandible, from the mandibular apex to the anterior clypeal margin.EL. *Eye length.* Maximum length of compound eye in lateral view [27].SL. *Scape length.* Maximum length of scape in dorsal view from apex to basal flange, not including basal condyle and neck [27].WL. *Weber’s length*. Diagonal length of mesosoma, from the anterior pronotal slope to the distal edge of the metapleura [44].PrW. *Pronotum width.* Maximum width of the pronotum in dorsal view [44].MFL. *Metafemur length* (most suitable view). Maximum length of the metafemur, measured from the distal margin of the trochanter to the metafemur apex [45].PL. *Petiole length*. Maximum length of the petiole in lateral view [44].PH. *Petiole height.* Maximum height of the petiole in lateral view [44].PPL. *Postpetiole length.* Maximum length of the postpetiole in lateral view.PPH. *Postpetiole height.* Maximum height of the postpetiole in lateral view.ATW. *Abdominal tergum IV width.* Maximum width of the fourth abdominal tergum with anterior, posterior, and lateral borders in the same plane of focus [46].

#### 4.2.3. Automontage Images 

We obtained high-resolution images using a Leica M205C magnifying stereoscope attached to a Leica DFC425 video camera at the MPEG. All illustrations were edited using Adobe Photoshop CS7^®^ for adjustments to enhance brightness and contrast details.

#### 4.2.4. Distribution Map 

The geographic coordinates obtained from specimen labels were entered/confirmed by Google Earth 7.1^®^ software, and distribution maps were created with Quantum GIS 2.18.15^®^ software. The shapefile of the Andean biogeographical region was obtained from published data [47].

### 4.3. Chemical Analysis

Gas chromatography–mass spectrometry (GC–MS) was performed on four workers and four larvae from two ant colonies (RMMA180623-18 and CRC180623-09) collected near El Llano, Panama (GPS). Single ants and two larvae were submerged and extracted whole in methanol. The GC–MS was set to EI mode, using a Shimadzu QP-2010 GC–MS or a Shimadzu QP-2020 GC–MS with an RTX-5, 30 m × 0.25 mm i.d. column. This instrument was run in the splitless mode and programmed from 60 to 250 °C at 10°/min with a flow rate of 1.5 mL/min.

### 4.4. Artificial Preparation of Alkaloid

2-Butyl-5-heptylpyrrolidine was prepared according to the method of Jones et al. [48], by the reductive amination of 5,8-pentadecadione to provide a cis/trans mixture of 2-butyl-5-heptylpyrrolidine. The synthetic alkaloid was diluted in molecular-grade ethanol to produce a stock solution concentration of 20,000 μg/mL, a concentration experimentally confirmed to inhibit bacterial growth by disk-diffusion assays (mean inhibition zone diameter: *E. coli* CSH36, 24.7 mm, and *S. saprophyticus* ATCC15305, 29.6 mm). A twofold serial dilution was performed on this stock solution to yield eight diluted concentrations, which were stored in glass vials at −80 °C.

### 4.5. Raw Venom Extraction

Raw venom extractions were conducted following the methods of Storey et al. [49] with minor modifications. *Megalomyrmex peetersi* venom solutions (*n* = 5; four from RMMA180623-18 and one from CRC180623-09) were prepared by extracting venom reservoirs from workers by pulling on the last two abdominal sclerites and manually removing the organ with forceps. Connected stingers and tissues were not removed to avoid a tear in the reservoir. For each solution, 20 venom reservoirs were placed in 200 μL of molecular-grade ethanol in a glass vial. Once all venom reservoirs were in solution, the contents of the glass vial were transferred to a plastic Safe-Lock tube and centrifuged at 8000 rpms for 10 min to push the venom out of the reservoir and into the ethanol solvent. The supernatant was then removed and placed back into the original glass vial, and the remaining tissue was discarded. Each sample was quantified using GC–MS techniques outlined in above sections.

### 4.6. Minimum Inhibitory Concentration (MIC) Assays

#### 4.6.1. Strains

The antimicrobial effects of 2-butyl-5-heptylpyrrolidine were evaluated using six bacterial strains: three Gram-positive (*Bacillus subtilis* NCIB3610, *Corynebacterium stationis* ATCC6872, *Staphylococcus saprophyticus* ATCC15305) and three Gram-negative (*Ralstonia pickettii* ATCC27511, *Aquaspirillum serpens* ATCC27050, *Escherichia coli* CSH36). *B. subtilis* NCIB3610 was obtained from the Bacillus Genetic Stock Center (BGSC; Columbus, OH, USA), and all other strains were obtained from the collections of the Department of Microbiology at the Ohio State University. Cultures were streaked on either Mueller–Hinton agar (BD BBL™ BD 211438, Franklin Lakes, NJ, USA) or tryptic soy agar (Thermo Scientific™ R455002, Waltham, MA, USA) and incubated between 26 and 37 °C.

#### 4.6.2. Determination of MIC

The MIC for each bacterial strain was determined using a broth microdilution procedure modified from the Clinical and Laboratory Standards Institute [50]. Overnight cultures were diluted to a 0.5 McFarland standard (1.0 × 10^8^ CFU/mL) with Mueller-Hinton broth (BD 275730), and then further diluted by a factor of five. An amount of 5 μL of the diluted overnight culture was placed into each well (excluding the blank wells for media control treatment), resulting in a final inoculum concentration of approximately 5.0 × 10^5^ CFU/mL.

Each row of a 96-well flat-bottomed microplate (Thermo Scientific™ 243656) was considered one experimental replicate. For each replicate, 5 μL of each alkaloid dilution was added to wells 1–8 in descending order of concentration, ranging from 500 to 3.90625 μg/mL. Each replicate also contained a well with 5 μL diluted tetracycline (Fisher BioReagents BP912-100), a well with 5 μL molecular-grade ethanol (negative solvent control), a growth control, and a blank control with only media. Due to the difficulties of diluting solid tetracycline in molecular-grade ethanol, we had to optimize the solution and dilute at 1300 μg/mL for a final well concentration of 32.5 μg/mL. This concentration is comparable to the maximum concentration of our synthetic alkaloid. Six replicates of the synthetic alkaloid across three plates and triplicates of the raw venom across three plates were used for each bacterial strain.

Each plate was placed into an ELx808i™ Absorbance Microplate Reader (BioTek Instruments, Winooski, VT, USA) and incubated at 37 °C for 24 h, with a plate shake and an absorbance reading at 600 nm (OD_600_) every 5 min for a 24 h period [51]. A total of 288 OD_600_ reads were taken for each well over 24 h, for a total of 2304 OD reads per treatment. OD_600_ reads from the unaltered growth treatment in column 11 of each trial were analyzed for the time at which cultures reached mid-log phase using the R [52] package Growthcurver v3.6.0 [53]. The OD_600_ for each bacterial strain per plate was determined for each treatment under their corresponding times at mid-log phase. Each bacterial strain was tested for significant differences in OD_600_ among treatments using a Kruskal–Wallis test. Dunn post hoc comparisons were used to determine the MIC of the synthetic alkaloid (the lowest concentration with a significant difference in OD_600_ from the negative solvent control, EtOH). 

### 4.7. Insecticidal Function of Raw Venom

To measure insecticidal function, we performed toxicity assays on *Reticulitermes flavipes* termites (see [40]) using raw venom from living ants. We sampled equally from two *Megalomyrmex peetersi* colonies (RMMA180623-18 and CRC180623-09) and five *Reticulitermes flavipes* termite colonies, creating a total of 10 replicates. For each ant–termite replicate, five ants and five termites were used. Before manipulation, the ants were anesthetized in petri dishes on ice. Then the ants were held by the petiole with forceps while the tip of the gaster was stroked with a microcapillary until the sting was everted and a droplet of venom formed. Droplets were applied to the termites on the head between the frons and mouthparts. This method mimics how *Megalomyrmex* ants apply their venom to prey items since they cannot inject their sting. We controlled for handling by holding the termites and touching their head with an empty microcapillary. A single ant was used for the application of venom on one termite. After being “stung,” the termites were placed inside 5 cm wide petri dishes lined with dry filter paper, which were then placed in a larger chamber lined with wet paper towel to prevent death by desiccation. Initial termite reactions were observed for 5 min, followed by a check after 1, 2, 4, and 6 h. The plates were tipped to the side upon every check. Standing individuals were recorded as “alive” if seemingly unaffected or “intoxicated” if moving erratically or frozen in place. Upside-down individuals were recorded as “incapacitated” if still moving or “dead” if not moving. Mortality was plotted and statistical significance assessed with pairwise log-rank comparisons from the R [52] package survminer v0.4.8 [54].

### 4.8. LD_50_ Assays of Synthetic Alkaloid

To measure alkaloid toxicity, we used synthesized alkaloid in assays similar to insecticidal assays with raw venom, but only measured mortality. We sampled equally from five colonies of *Reticulitermes flavipes* so that individuals from each colony were exposed to all alkaloid dilutions. An amount 0.5 μL of each alkaloid dilution was applied to the termites on the head between the clypeus and frons. For each dilution–termite replicate, five individuals were used. Following alkaloid administration, the termites were placed inside 5 cm wide petri dishes lined with dry filter paper, which were then placed in a larger chamber lined with wet paper towel to prevent death by desiccation. Initial termite reactions were observed for 5 min, followed by a check after 1, 2, 4, and 6 h. The plates were tipped to the side upon every check. The individuals were recorded as “dead” if upside down and immobile. Mean weight was measured independently using 20 termites from two colonies. LD_50_ after 6 h was calculated from a four-parameter log-logistic function using the R [52] package drc v3.0-1 [55].

## Figures and Tables

**Figure 1 toxins-12-00679-f001:**
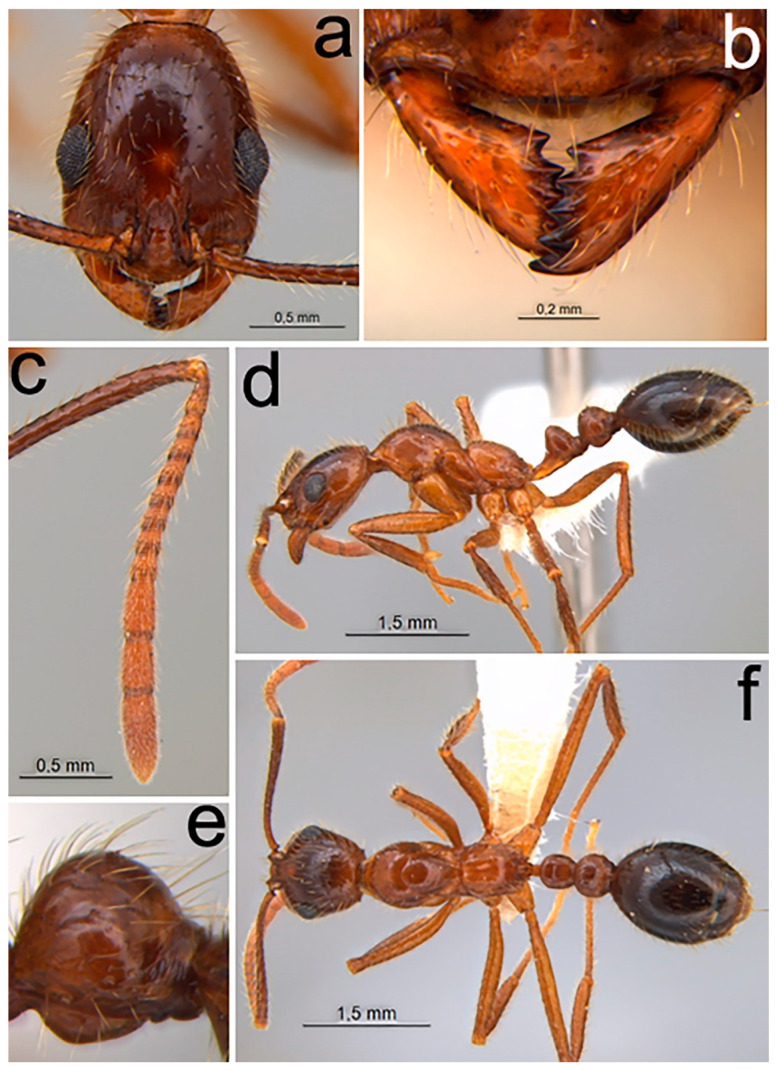
*Megalomyrmex peetersi* sp. n. holotype worker. (**a**) Head in frontal view, (**b**) mandibles, (**c**) funiculus of antenna, (**d**) lateral view, (**e**) postpetiole, in lateral view, and (**f**) dorsal view.

**Figure 2 toxins-12-00679-f002:**
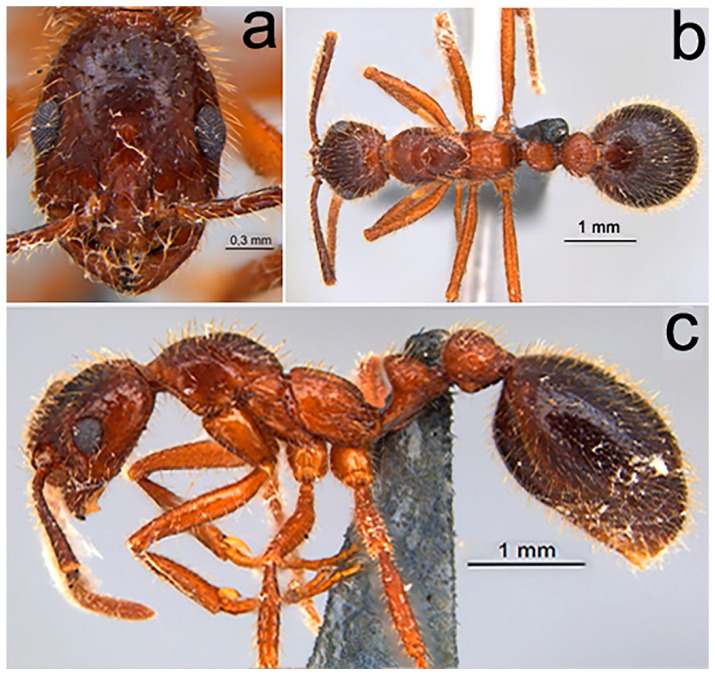
*Megalomyrmex peetersi* sp. n. ergatoid queen. (**a**) Head in frontal view, (**b**) dorsal view, and (**c**) lateral view.

**Figure 3 toxins-12-00679-f003:**
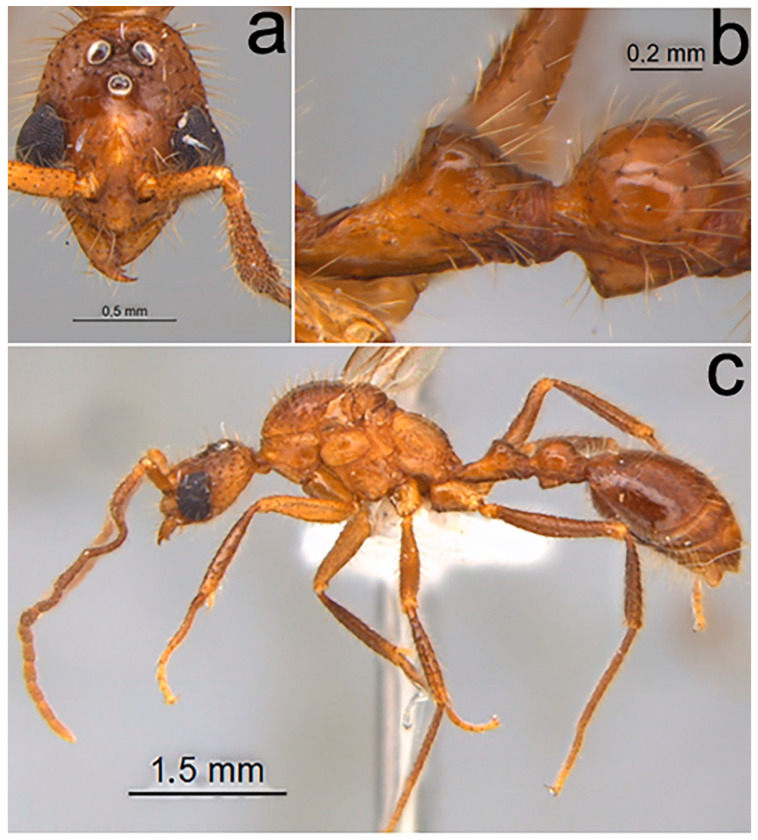
*Megalomyrmex peetersi* sp. n. male. (**a**) Head in frontal view, (**b**) waist, in lateral view, and (**c**) lateral view.

**Figure 4 toxins-12-00679-f004:**
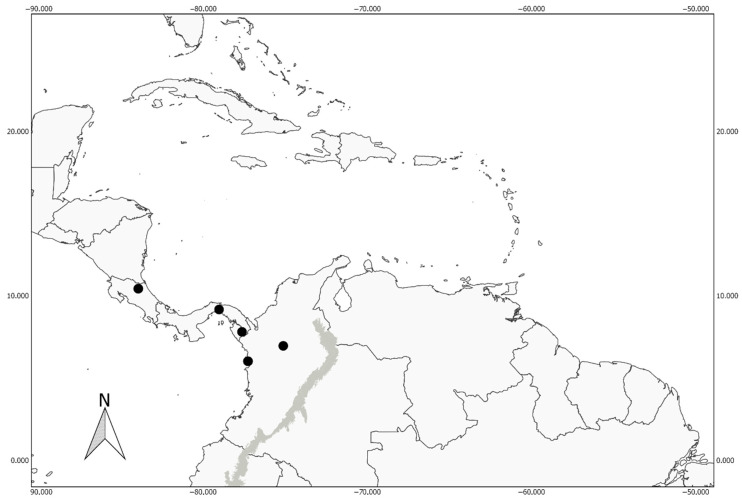
Distribution map of *Megalomyrmex peetersi* sp. n. indicated with solid dot.

**Figure 5 toxins-12-00679-f005:**
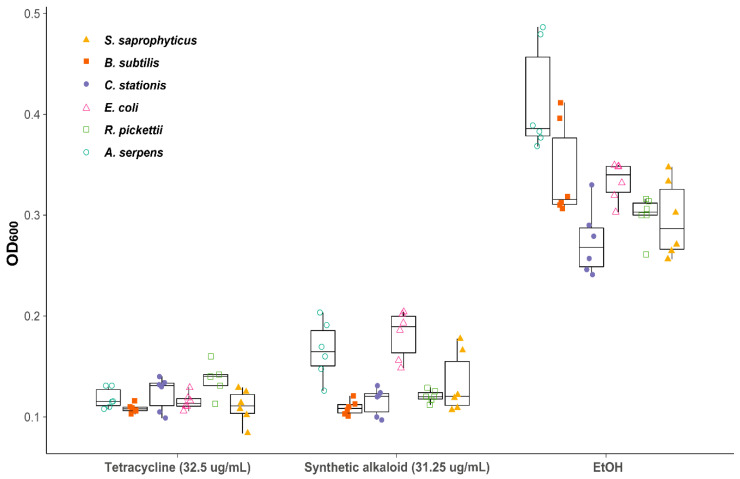
OD^600^ at mid-log phase for six bacterial strains treated with 2-butyl-5-heptylpyrrolidine at a concentration of 31.25 μg/mL. EtOH was used as a negative solvent control, and tetracycline was used as a positive control. Filled shapes indicate Gram-positive strains and hollow shapes indicate Gram-negative strains.

**Figure 6 toxins-12-00679-f006:**
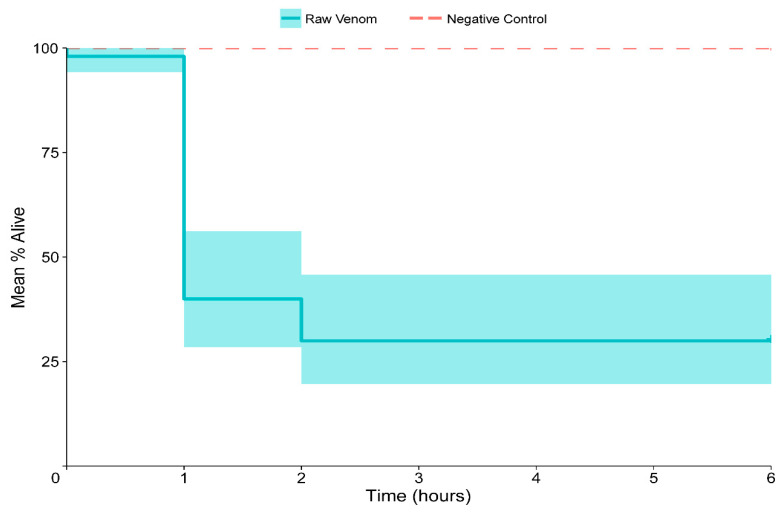
Toxicity of *Megalomyrmex peetersi* sp. n. venom in termites over time. The solid bar shows mortality with a lighter buffer representing standard deviation. The dashed bar shows an ethanol control. Compared with the control, *M. peetersi* sp. n. results were significantly different (*p* < 0.01).

**Table 1 toxins-12-00679-t001:** Minimum inhibitory concentration (MIC) of 2-butyl-5-heptyl-5-pyrrolidine tested against six bacterial strains.

Bacterial Strain	MIC (μg/mL)
*Escherichia coli*	62.5
*Staphylococcus saprophyticus*	62.5
*Bacillus subtilis*	62.5
*Aquaspirillum serpens*	62.5
*Corynebacterium stationis*	31.25
*Ralstonia pickettii*	62.5

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
