# Peer review of "Venom Function of a New Species of Megalomyrmex Forel, 1885 (Hymenoptera: Formicidae)"

_toxins, 2020, doi:10.3390/toxins12110679_

Round 1

Reviewer 1 Report

The research presented in the paper is interesting and, for the most part, thorough. The authors demonstrate the isolation of an alkaloid toxin from the venom of a new species of Megalomyrmex, and offer compelling evidence of its antibacterial (need clarification whether cytotoxic?) and insecticidal properties. The introduction is comprehensive and provides interesting background of related venomous ant species. I request some clarification or moderate changes to the following line items, and afterwards would like to express some concerns I have regarding the MIC assays:

42) I don't understand the phrasing. Is some kind of complex being formed, or is the nature of the interspecific competition being described as complex? 

53 and 79) be consistent with gaster flagging/gaster-flagging being hyphenated or not 

57) do references 22-26 also pertain to this statement? reference 21 is unpublished, and the next paragraph opens with discussing structural diversity and describing 5 classes of alkaloids. 

74) references should be stated in ascending order, [21,25]

333-334) how were ants transported? Were the same enclosures that were used for storage used for transport?

393-394) Please include the flow rate for the GC-MS studies. Also indicate whether the method was split or splitless

400) is there a reference for the concentrations you decided on using for the disk diffusion assays, or is this something you determined through your own experimentation?

411-412) it is standard practice to filter liquids before injecting onto a GC-MS (particularly when extracting from organisms such as insects). Was there any filtering implemented during processing/ before injection? 

In addition to the line items mentioned, I have a broader concern regarding the antimicrobial studies. It seems the use of 25 ug/mL final concentration of tetracycline is arbitrary. I would like to see what the MIC of your strains are for tetracycline if you are going to claim that the compound of interest is of similar efficacy to tetracycline. The experiment would be improved with exact MIC comparisons rather than a statement of comparative efficacy without data. It's also peculiar that you didn't include a fungus in your studies (perhaps Candida or Saccharomyces), as this would indicate whether the compound is truly antibacterial or cytotoxic. It is important to show that the compound is an antibacterial and not just a potent cellular toxin that kills everything. 

Reviewer 2 Report

The authors present a comprehensive investigation wherein they isolate and characterize an alkaloid compound obtained from the venom of a newly discovered species of ant, Megalomyrmex peetersi sp. that is free living.  As the authors note, the use of alkaloids is generally toxic to ants, with only 10 of 500 genera found to utilize these compounds.  The genus of interest has been found to produce five classes of alkaloids: pyrrolidines, pyrrolizidines, piperidines, pyrrolines, and indolizidines.  The authors specifically focus on an exclusive venom alkaloid used by these ants to repel other ants they present as trans-2-butyl-5-heptylpyrrolidine.

In meticulous and beautiful detail, the authors first describe the physical characteristics of the various types (e.g., workers, queen) of ant of this species via detailed photographs and biometrics.  The locations the specimens were collected from are provided, and the explanation for the new species name provided for historical value.

The isolation and biochemical characterization of the alkaloid of interest is presented well, and the antibacterial studies with comparison to tetracycline and EtOH (as negative control) very convincing.  They identified and then manufactured the alkaloid of interest and compared it to the effects of the raw venom, finding similar results in the systems utilized. 

It is interesting that while the raw venom tended to inhibit bacterial growth, it was not as effective as the alkaloid compound – perhaps because there was an interaction with the venom or ETOH, or perhaps because the ETOH was not quite as negative a control as the authors had hoped.  this should be addressed.

The further in vivo effects on termites to show the toxicity and ultimately lethality of the compound is also well done.  I appreciate that this is a multi-year, multidisciplinary work.  Please explain in the legend of figure 6 what “variation” designated by the pale blue color means (SEM, SD, etc.).

The discussion is fair, with details concerning how the ants likely apply the venom/alkaloid to their brood for protection is very thoughtful and detailed.  The use of the venom as an evolutionary advantage as espoused by the authors.  Lastly, the limitations of the investigation are fairly presented, and the conclusions drawn reasonable.
